# The Anti-Inflammatory and Skin Barrier Function Recovery Effects of *Schisandra chinensis* in Mice with Atopic Dermatitis

**DOI:** 10.3390/medicina59071353

**Published:** 2023-07-24

**Authors:** Yoorae Son, Wonjin Yang, Sangjun Park, Jinkyu Yang, Soyeon Kim, Ji-Hyo Lyu, Hyungwoo Kim

**Affiliations:** 1Division of Pharmacology, School of Korean Medicine, Pusan National University, Yangsan 50612, Republic of Korea; yoorae58@naver.com (Y.S.); haniyang1@naver.com (W.Y.); sjpark177@gmail.com (S.P.); vo2mxlift@gmail.com (J.Y.); amugdale@pusan.ac.kr (S.K.); 2Herbal Medicine Resources Research Center, Korea Institute of Oriental Medicine, Naju 58245, Republic of Korea

**Keywords:** *Schisandra chinensis*, herbal medicine, inflammation, atopic dermatitis, eczema

## Abstract

*Background and Objectives*: The fruit of *Schisandra chinensis* (Turcz.) Baill. is widely used medicinally to treat coughs, asthma, exhaustion, eczema, and pruritus in Northeast Asian countries, including Korea, China, and Japan. This study was designed to investigate the effects of *S. chinensis* on dermatitis in mice with calcipotriol (MC-903)-induced atopic dermatitis (AD), and its effects on skin barrier dysfunction was also investigated. *Materials and Methods*: The inhibitory effects of an ethanolic extract of *S. chinensis* (EESC) on skin lesions, water content, water-holding capacity (WHC), histopathological abnormalities, and inflammatory cytokine and chemokine levels were evaluated in mice with AD induced by MC903. *Results*: Topical EESC ameliorated skin lesions, reduced skin water content, and increased MC903-induced WHC. EESC also prevented MC-903-induced histopathological abnormalities such as epidermal disruption, hyperkeratosis, spongiotic changes, and immune cell infiltration in inflamed tissue. Moreover, topical EESC reduced MC-903-induced levels of pro-inflammatory cytokines and chemokines, such as tumor necrosis factor (TNF)-α, interleukin (IL)-1β, IL-4, IL-6, IL-8, monocyte chemotactic protein (MCP)-1, and thymic stromal lymphopoietin (TSLP). Furthermore, unlike dexamethasone, EESC did not reduce the spleen/body weight ratio. *Conclusions*: These results suggest that *S. chinensis* can be used as an alternative to external corticosteroids and that its anti-inflammatory and skin barrier dysfunction-restoring effects are related to the downregulation of pro-inflammatory cytokines and chemokines, such as TNF-α, IL-4, IL-6, IL-8, and TSLP.

## 1. Introduction

The fruit of *Schisandra chinensis* (Turcz.) Baill. (Omija in Korea, Wǔwèizǐ in China) is a traditional herbal medicine with a sour taste and warm properties used to promote the production of body fluid, and has been widely used in Northeast Asian countries, including Korea, China, and Japan, to treat coughs, asthma, exhaustion, eczema, and pruritus [1]. Recently, interest in *S. chinensis* and its active ingredients as raw materials for skin products has increased. It has been reported that *S. chinensis* contains ingredients with antioxidant effects, so it has a skin-whitening effect and suppresses skin irritation [2,3]. In addition, *S. chinensis* has been reported to have anti-oxidative [2], anti-aging [4], anti-allergic [5], and anti-inflammatory [6] effects, and the two primary components of *S. chinensis*, α-cubebenoate and gomisin M2, have been found to have anti-inflammatory effects in mice with experimental AD [7,8,9].

Because the manifestations of atopic dermatitis (AD) are diverse and its differential diagnosis is broad, it may not be possible to treat the condition using a single targeted treatment [10]. Therapists manage atopic patients using various combinatorial treatment methods, including natural products, baths, moisturizing creams, and topical steroids, based on the assessment of an individuals’ symptoms [11]. Among them, natural products such as plant extracts are extensively studied as complementary alternatives to corticosteroids [12].

AD causes various skin barrier function abnormalities, such as destruction of stratum corneum [13], changes in lipid composition, reductions in natural moisturizing factor levels [14], and increased sensitivity to various triggers, with accompanying inflammation and itching [13,14]. Stratum corneum damage causes water loss, flare, irritation, and hypersensitivity associated with skin barrier function impairment, a causative factor and major symptom of AD and other chronic skin diseases [15].

Itching caused by dry skin is a major symptom of AD. Impaired skin barrier function results in increased skin permeability, dryness, immune activation, and nerve fiber stimulation, all of which can trigger itching sensations [16]. Managing pruritus in conditions like AD involves addressing skin barrier dysfunction, reducing inflammation, and employing itch management techniques [17].

We investigated the anti-inflammatory and skin barrier function healing effects of *S. chinensis* fruit in an MC903 (calcipotriol, a low-calcemic analogue of vitamin D3)-induced mouse model of AD using a lesion severity scoring system, skin water content, cytokine levels, and histopathological findings.

## 2. Materials and Methods

### 2.1. Preparation of the Schisandra Chinensis Extract

*S. chinensis* fruit was purchased from the Kwangmyungdang Drug Company (Ulsan, Korea) and certified by Professor Hyungwoo Kim (Pusan National University) (voucher no. MS2022-1001). Chopped fruits were extracted using 100% ethanol to produce an ethanol extract of *S. chinensis* (EESC), as previously described (yield, 15.28%) [18].

### 2.2. Animals

The animal study was conducted using Balb/c mice (male, 8-week-old, Hana Biotech, Gyeonggi, Republic of Korea). The mice were housed in a specific pathogen-free environment under a 12 h light/dark cycle and provided standard rodent food and water *ad libitum*. All animal experiments were conducted in accordance with the guidelines issued by the Animal Care and Use Committee of Pusan National University (PNU-2019-2269).

### 2.3. Experimental Design and the Induction of AD

Experimental AD was induced using a slight modification of the method described by Martinez et al. [19]. Briefly, mice were randomly divided into six groups. MC903 was diluted to 0.1 mM and administered to shaved dorsa on eight consecutive days (days 4–11) at 4 nmol/day to animals in the five experimental groups to induce AD. Animals in the NOR (normal) group were treated with vehicle for eight days (n = 6). The AD controls (CTL) were treated with MC903 followed by the vehicle (n = 8). EESC-treated animals were treated with MC903 followed by EESC (60, 180, or 600 μg/day) for six consecutive days (n = 8/group). Dexamethasone (DEX; the positive control) was applied topically at 150 μg/day for six consecutive days (days 6–11). The experimental process is summarized in the Appendix A.

### 2.4. Assessment of Skin Lesion Severity

Skin lesion (erythema, scale, induration and fissure) severity was assessed on day 12 using a four-point scale (0 = no, 1 = slight, 2 = mild, and 3 = severe symptoms), as previously described [18].

### 2.5. Skin Thicknesses and Color

Skin thickness was measured using a Vernier caliper (Mitutoyo, Tokyo, Japan). Dorsal skin color was evaluated on day 12 at three different locations per mouse using a skin colorimeter (DSM II, Cortex Technology, Aalborg, Denmark). Three measures were taken per mouse and averaged, and these averages were used to calculate group averages.

### 2.6. Skin Water Content and Water-Holding Capacity

The effects of EESC on skin water content and water-holding capacity (WHC) were determined using a skin hygrometer (Scalar Corporation, Tokyo, Japan). Water content was measured three times using different locations per mouse, and averages were calculated. For WHC measurements, a square piece of gauze (1 × 1 cm) soaked in distilled water (DW) was placed on the shaved back of a mouse and immediately removed. Skin moisture content was measured at three different locations every 30 s.

### 2.7. Histopathological Examination

Inflamed tissue was dissected, embedded in paraffin, stained with hematoxylin and eosin (H&E), and observed under an optical microscope (100×). Severity was evaluated semi-quantitatively using a four-point scale to assess the degree of hyperkeratosis, spongiotic changes, and angiogenesis. The extent of immune cell infiltration was assessed as previously described [18].

### 2.8. Total RNA Isolation and Quantitative PCR

Total RNA was isolated from dorsal skin tissue using Trizol (Invitrogen, Carlsbad, CA, USA). Complementary DNA (cDNA) was synthesized as previously described [20], and quantitative PCR (qPCR) was performed using a Rotor-Gene Q unit (Qiagen, Hilden, NRW, Germany) and TOPreal SYBR Green qPCR premix (Enzynomics, Daejeon, Korea). GADPH was used as the housekeeping gene, and threshold values (Ct values) were normalized using the 2 − ΔCt formula for quantitative analysis. The primer sets used in this study are provided in Appendix A.

### 2.9. Cytokine Level Measurements

Cytokine levels in skin samples were assessed using a cytometric bead array mouse inflammation kit (BD, San Jose, CA, USA) [18]. 

### 2.10. Body Weight and Spleen/Body Weight Ratio

Changes in body weight between days 1 and 12 are expressed as percentages of body weight on day 1. Spleen/body weight ratios were calculated using body and spleen weight on day 12.

### 2.11. Statistical Analysis

The significances of intergroup differences were determined by one-way ANOVA and Dunnett’s multiple comparison test. The analysis was performed using Prism 5 (version 5.01) software. The results are expressed as means ± standard deviations (SD), and statistical significance was accepted for *p*-values < 0.05.

## 3. Results

### 3.1. EESC Relieved Skin Lesions and Reduced the Melanin Index

Repeated application of MC903 to shaved dorsal skin caused induration, fissure, scale, erythema, and petechiae. Topical treatment with EESC for six consecutive days reduced these symptoms (Figure 1A). EESC (600 μg/day) and DEX significantly reduced severity scores and melanin indices compared to the CTL group (Figure 1B,D). 

### 3.2. EESC Improved the Skin Water Content and Water-Holding Capacity

Marked decreases in water content were observed in the CTL group. Topical application of EESC significantly elevated the water content compared to the CTL group (Figure 2A). In the CTL group, a significant decrease in water content was noted 60 and 90 s after the start of the experiment, and a significant increase in WHC was observed in the 180 and 600 μg/day EESC-administered groups compared to the CTL group (Figure 2B). DEX had no significant effect on skin water content or WHC (Figure 2).

### 3.3. EESC Prevented Histopathological Abnormalities in Inflamed Tissue

Topical application of MC903 disrupted the basal cell layer and produced an indistinct boundary between the epidermis and dermis, spongiotic changes, and immune cell infiltration (Figure 3A). EESC treatment at 600 μg/day significantly reduced the inflamed tissue severity score by 51% and immune cell infiltration by 43% compared to the CTL group (Figure 3B,C). DEX also significantly reduced the severity scores and immune cell infiltration versus the CTL group (Figure 3).

### 3.4. EESC Suppressed MC-903-Induced Increases in Cytokine and Chemokine mRNA Levels

Marked increases in the expression of interleukin (IL)-1β, IL-4, IL-6, IL-8, thymic stromal lymphopoietin (TSLP), and monocyte chemotactic protein (MCP)-1 mRNA were observed in the CTL group. These increases were significantly suppressed by EESC (Figure 4).

### 3.5. EESC Suppressed the MC-903-Induced Production of Tumor Necrosis Factor (TNF)-α, IL-4, and IL-6 in Inflamed Tissue

Significant increases in cytokine (TNF-α, IL-4, IL-6, and IL-2) levels were observed in the CTL group. The topical application of EESC at 600 μg/day or DEX at 150 µg/day significantly decreased TNF-α, IL-4, and IL-6 levels (Figure 5). 

### 3.6. EESC Did Not Affect the Weight Gain or Spleen/Body Weight Ratio in AD Mice

AD induction significantly decreased the average body weight. EESC and DEX did not affect body weight (Figure 6A). However, DEX significantly reduced the spleen/body weight ratio compared to the CTL and NOR controls (Figure 6B).

## 4. Discussion

In Northeast Asian traditional medicine, the cause of skin dryness or itching is attributed to “wind heat” caused by a yin deficiency. Based on this theory, traditional medicine experts in Northeast Asia use methods that expel wind heat or supplement yin to treat skin diseases. *S. chinensis* is an important lung astringent, nourishes the kidneys, promotes body fluid production, and reduces perspiration. In addition, it also astringes the essence, nourishes the heart, and calms the mind [1]. Because of these effects, *S. chinensis* is used to treat dry or itchy skin caused by a lack of yin.

Recently, Martinez et al. [19] and Mei et al. [21] proposed the use of 1,25-(OH)2D3 (vitamin D3) and its derivative, calcipotriol (MC903), to produce animal models of AD. MC903 produces skin symptoms and inflammation in mice commonly observed in acute AD patients and increases serum IgE levels; thus, MC-903-induced animal models are an attractive option as they do not require the use of harmful substances such as oxazolone or 2,4-dinitrofluorobenzene (DNFB) [19]. 

In the current study, we established an MC-903-induced murine model of AD. Repeated application of MC903 to dorsal skin induced induration, fissure, scale, erythema, and petechiae and reduced skin water content and WHC, which are hallmarks of skin barrier dysfunction. These results indicate that our animal model well mimicked human AD.

As shown in Figure 1 and Figure 2, topical application of EESC significantly suppressed MC-903-induced skin lesions and reductions in skin moisture contents and WHC. These results imply that EESC can ameliorate AD symptoms by reducing the skin dryness caused by skin barrier dysfunction. 

Natural moisturizing factors (NMFs) such as filaggrin, loricrin, and involucrin keep the skin moist and reduce sensitivity [22]. In previous studies, an ethanolic extract of *S. chinensis* significantly restored the reduction in filaggrin expression by IL-4 plus IL-13 stimulation in HaCaT keratinocytes [23] and gomisin N, one of active compounds of *S. chinensis,* induces expression of the ceramide synthesis gene and reduces the ceramide degradation gene in HepG2 cells [24]. These results and our findings suggest that *S. chinensis* has the potential to restore skin barrier dysfunction in AD animal models by restoring impaired NMF such as filaggrin and ceramide.

In addition, active compounds of *S. chinensis* are known to have tyrosinase inhibitory activity and melanin production inhibitory effects in cells [25], and *S. chinensis* has been proposed as a skin-whitening agent [26]. Considering previous results, the decrease in melanin index (Figure 1E) suggests its potential use as a skin-whitening agent.

Histopathological findings in AD patients depend on disease stage, severity, and patient characteristics. Common histopathological findings of AD include hyperkeratosis, spongiosis, exocytosis, epidermal microabscesses, immune cell infiltration, fibrosis, and angiogenesis [27]. We found that EESC effectively suppressed MC-903-induced indistinct epidermis/dermis boundary formation caused by basal cell layer disruption, epidermal tissue damage, spongiotic changes, hyperkeratosis, and immune cell infiltration (Figure 3A). These results indicate that EESC has an anti-inflammatory effect and prevents MC-903-induced skin lesion formation and skin barrier dysfunction.

In addition, α-cubebenoate and gomisin M2 isolated from *S. chinensis* prevented eosinophil and mast cell infiltration in AD mice and prevented antigen-induced degranulation in RBL-2H3 cells [8,9]. These results suggest that the effect of suppressing immune cell infiltration of *S. chinensis* can be explained by the above components.

Epidermal thickening (acanthocytosis) is often observed in patients with chronic AD, and epidermal deterioration manifesting as epidermal thinning, cracking, or erosion directly causes skin barrier dysfunction and susceptibility to secondary infections in patients with severe AD [27]. In our animal model, epidermal deterioration and skin barrier dysfunction were both observed in the CTL group, and EESC effectively suppressed these abnormalities. These observations indicate that our animal model is one of moderate-to-severe AD and that EESC suppresses MC-903-induced skin barrier dysfunction by inhibiting epidermal deterioration.

TNF-α and IL-1β are pro-inflammatory cytokines that play crucial roles in the immune response and can contribute to the pathogeneses of various inflammatory conditions, including AD [28]. TNF-α can contribute to skin barrier disruption by impairing the expression of proteins involved in skin barrier function and promoting epidermal hyperplasia and inflammation [29]. IL-1β can induce epidermal hyperplasia, disrupt skin barrier function, and promote the release of other pro-inflammatory cytokines [29]. We found that EESC effectively inhibited the production of TNF-α and IL-1β (Figure 4 and Figure 5), indicating that EESC might restore epidermal disruption and skin barrier dysfunction by inhibiting pro-inflammatory cytokines such as TNF-α and IL-1β.

Th2-skewing cytokines such as IL-4, IL-5, IL-6 and TSLP participate in the pathogenesis of AD and other allergic conditions. IL-4 is a cytokine required for Th2 cell differentiation, IgE production, and eosinophil recruitment [30], whereas IL-6 induces the production of other pro-inflammatory cytokines, including IL-1β and TNF-α, by keratinocytes and modulates the production and distribution of key proteins involved in skin barrier integrity, such as filaggrin and tight junction proteins [31]. We found that EESC effectively inhibited the MC903-induced production of IL-4 and IL-6 (Figure 4 and Figure 5). These results indicate that suppression of skin barrier dysfunction by EESC is related to the suppression of Th2-skewing cytokines.

IL-8 (also known as chemokine (C–X–C motif) ligand 8, CXCL8) is a chemokine that plays a significant role in inflammation and immune responses [28]. In addition, MCP-1 (also known as chemokine (C–C motif) ligand 2, CCL2) functions as a chemoattractant for various immune cells, including monocytes, T cells, neutrophils, and dendritic cells [32]. Furthermore, IL-6, TNF-α, and interferon (IFN)-γ promote immune cell recruitment and inflammation by enhancing the expressions of adhesion molecules in keratinocytes [33,34]. In the present study, EESC significantly reduced MC903-induced immune cell infiltration into perivascular areas (Figure 3C) and inhibited the production of IL-6, IL-8, and MCP-1 (Figure 4 and Figure 5). These results indicate that EESC can reduce immune cell infiltration by suppressing the secretion of several chemokines.

TSLP plays a multifaceted role in AD and contributes to skin barrier dysfunction, immune dysregulation, and pruritus [28]. Additionally, its ability to disrupt epidermal differentiation, impair lipid barrier function, and influence immune responses contribute to the development and progression of AD [35]. Our observation that EESC inhibited MC903-induced TSLP production (Figure 4F) is encouraging, as TSLP modulates key therapeutic targets related to skin barrier dysfunction and excessive Th2 immune response.

MC903 (calcipotriol) is a ligand of vitamin D receptor (VDR)-like receptors that can induce TSLP secretion [36], and it has been reported that TSLP expression is mediated via the retinoic acid receptor γ1 (RARγ) and retinoid X receptor (RXR) pathways. Furthermore, synthetic agonists of VDR and RARγ and the natural agonist all-trans retinoic acid (ATRA) increase TSLP expression in the skin [37]. 

Among the components of *S. chinensis* detailed in the TCMSP (Traditional Chinese Medicine Systems Pharmacology Database and Analysis Platform) database, 21, including gomisin A, G, and R, meet the oral bioavailability (OB, ≥20%) and drug likeness (DL, ≥0.1) outlined by the ADME (absorption, distribution, metabolism, and excretion) criteria (Appendix A). In addition, 45 targets, including PTGS2 (prostaglandin G/H synthase 2, COX-2) and GABRA1 (gamma-aminobutyric acid receptor subunit alpha-1), were identified, and these targets interacted with nine active components (Appendix A). Correlations between four inflammation-related targets (PTGS1, PTGS2, NOS3, and RXRA) and active components of *S. chinensis* were analyzed using Cytoscape 3.91, and interestingly, dibutyl phthalate and deoxyshikonin were found to interact with all targets (Appendix A). 

Some active components of *S. chinensis,* such as deoxyshikonin and dibutyl phthalate, inhibit nitric oxide and prostaglandin-related pathways and thus may be involved in the anti-inflammatory effect of *S. chinensis*. In addition, deoxyshikonin and dibutyl phthalate interact with RXRA and thus inhibit TSLP secretion by modulating the function of RXR, thereby reducing the inflammatory response and ameliorating skin barrier dysfunction.

According to Kang and Shin [38], a water extract of *S. chinensis* alleviated the symptoms of 1-chloro-2,4-dinitrobenzene (DNCB)-induced AD in NC/Nga mice, and Lee et al. [39] reported that a methanol extract of *S. chinensis* prevented ear swelling and suppressed cytokine and chemokine (TNF-α, INF-γ, IL-6, and MCP-1) levels in a Balb/c mouse DNFB-induced contact dermatitis model. These results are consistent with our results and suggest that *S. chinensis* extracts can inhibit skin inflammation.

Corticosteroids are fundamental for the treatment of AD and are commonly continuously administered when allergens cannot be avoided, such as in patients with occupational skin disease. In the present study, dexamethasone treatment reduced spleen size, an indicator of immune weakness, but EESC had no significant effect on spleen/body weight ratios, which suggests that EESC and corticosteroids act via different mechanisms and that EESC may not have systemic side effects, such as general immune-suppression.

## 5. Conclusions

In this study, we investigated the effects of an ethanol extract of *S. chinensis* on MC903-induced dermatitis and skin barrier dysfunction. EESC was found to suppress the MC903-induced production of pro-inflammatory cytokines (TNF-α and IL-1β) and Th2-skewing cytokines (IL-4, IL-6, and TSLP), and to attenuate inflammatory responses and epidermal disruption by MC903. In addition, EESC prevented MC903-induced immune cell infiltration by suppressing the expressions of chemokines (IL-8 and MCP-1). Furthermore, EESC suppressed MC903-induced histopathological abnormalities by suppressing cytokine and chemokine expressions and the severities of AD lesions and skin barrier dysfunction. Taken together, our findings suggest that *S. chinensis* can be used as an alternative to external corticosteroid administration, and that the anti-inflammatory and skin barrier protective effects of *S. chinensis* are related to the inhibition of cytokines and chemokines such as TNF-α, IL-4, IL-6, IL-8, and TSLP.

## Figures and Tables

**Figure 1 medicina-59-01353-f001:**
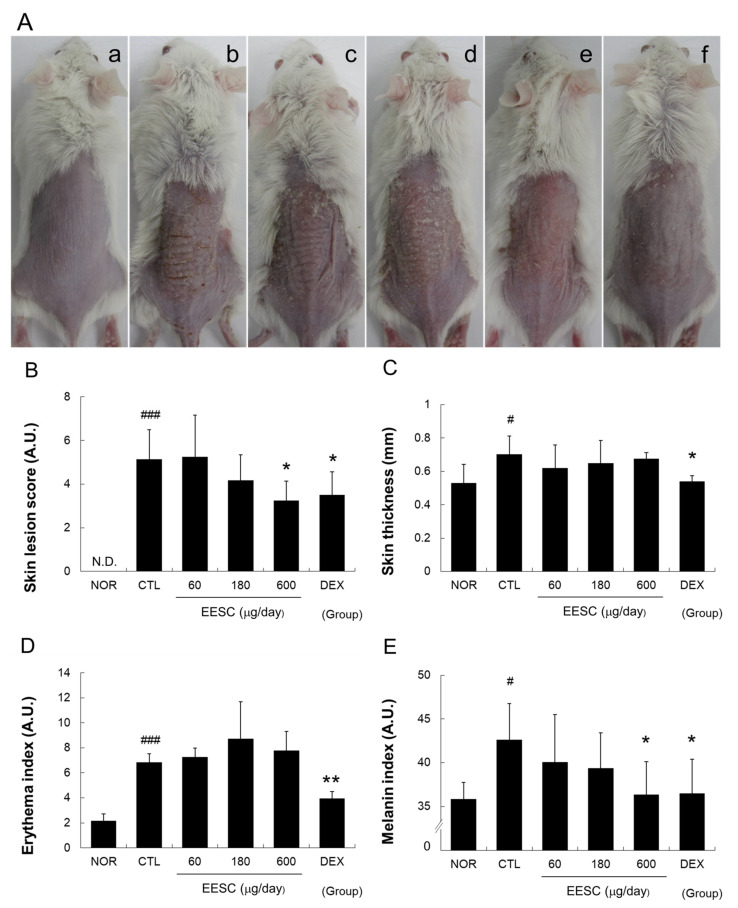
Effects of EESC on skin lesions and color in AD mice. (**A**) (**a**) Non-treated naïve (NOR); (**b**) AD control (CTL); (**c**) 60 μg/day of EESC; (**d**) 180 μg/day of EESC; (**e**) 600 μg/day of EESC; (**f**) 150 μg/day of DEX. (**B**) Skin lesion scores. (**C**) Skin thicknesses. (**D**) Erythema indices. (**E**) Melanin indices. A.U., arbitrary unit; N.D., undetectable; EESC, ethanol extract of *S. chinensis*; DEX, dexamethasone. Results are presented as means ± SDs. # *p* < 0.05 and ### *p* < 0.001 vs. the NOR group; * *p* < 0.05 and ** *p* < 0.01 vs. the CTL group.

**Figure 2 medicina-59-01353-f002:**
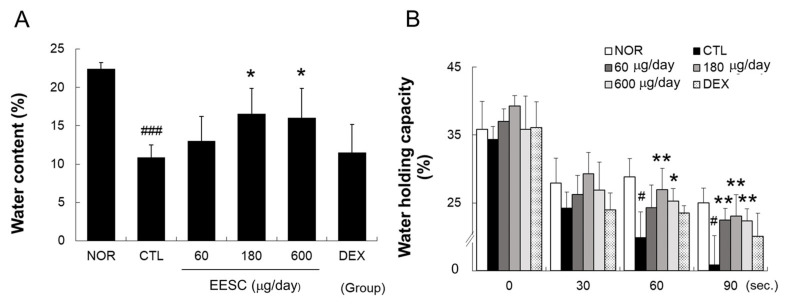
Effects of EESC on skin water content and WHC in AD mice. (**A**) Water content; (**B**) WHC. EESC, ethanol extract of *S. chinensis*; DEX, dexamethasone. Results are presented as means ± SD. # *p* < 0.05 and ### *p* < 0.001 vs. the NOR group; * *p* < 0.05 and ** *p* < 0.01 vs. the CTL group.

**Figure 3 medicina-59-01353-f003:**
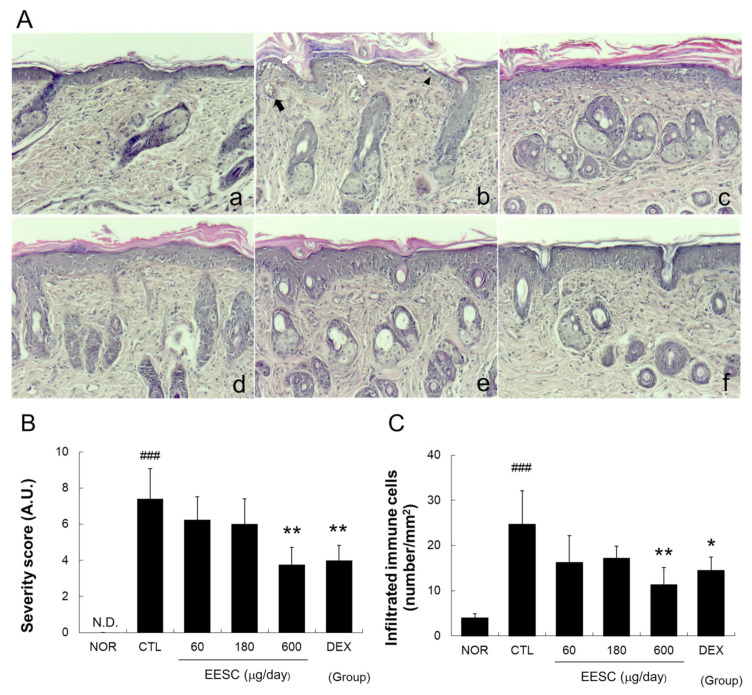
Effects of EESC on histopathological abnormalities in inflamed tissue. (**A**) The abbreviations used are the same as those used in Figure 1A. The solid arrow indicates a blood vessel near the epidermis. White arrows show epidermis disruption. The solid wedge indicates an indistinct boundary between the epidermis and dermis. (**B**) Severity scores; (**C**) infiltrated immune cells. N.D., undetectable.; EESC, ethanol extract of *S. chinensis*; DEX, dexamethasone. Results are presented as means ± SDs. ### *p* < 0.001 vs. the NOR group; * *p* < 0.05 and ** *p* < 0.01 vs. the CTL group.

**Figure 4 medicina-59-01353-f004:**
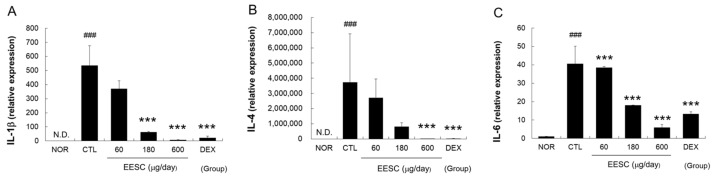
Effects of EESC on MC-903-induced increases in cytokine and chemokine mRNA levels in skin tissue. (**A**) IL-1β; (**B**) IL-4; (**C**) IL-6; (**D**) IL-8; (**E**) MCP-1; (**F**) TSLP. N.D., undetectable. EESC, ethanol extract of *S. chinensis*; DEX, dexamethasone. Results are presented as means ± SDs. ### *p* < 0.001 vs. the NOR group; * *p* < 0.05, ** *p* < 0.01 and *** *p* < 0.001 vs. the CTL group.

**Figure 5 medicina-59-01353-f005:**
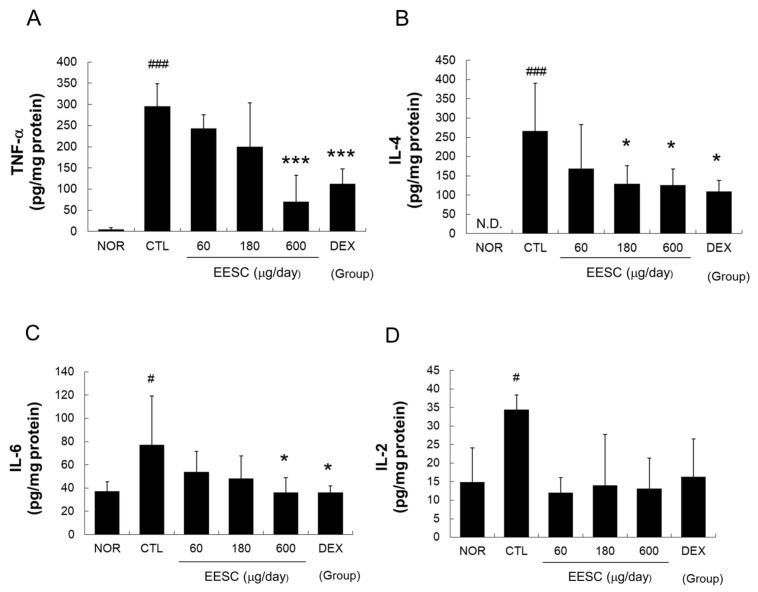
Effects of EESC on the MC-903-induced production of TNF-α, IL-4, IL-6, and IL-2 in skin tissue. (**A**) TNF-α; (**B**) IL-4; (**C**) IL-6; (**D**) IL-2. N.D., undetectable. EESC, ethanol extract of *S. chinensis*; DEX, dexamethasone. Results are presented as means ± SDs. # *p* < 0.05 and ### *p* < 0.001 vs. the NOR group; * *p* < 0.05 and *** *p* < 0.001 vs. the CTL group.

**Figure 6 medicina-59-01353-f006:**
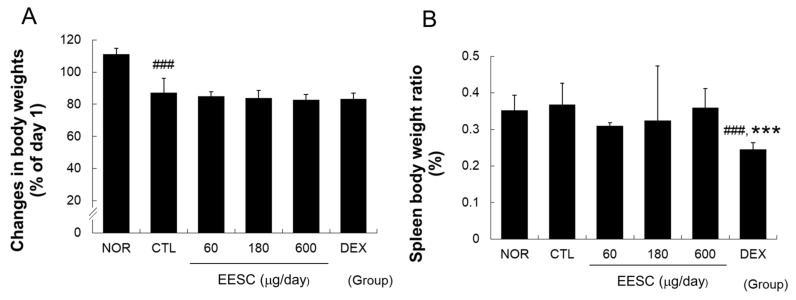
Effects of EESC on MC903-induced changes in body weight and spleen/body ratio. (**A**) Body weight; (**B**), spleen/body weight ratios. EESC, ethanol extract of *S. chinensis*; DEX, dexamethasone. Results are presented as means ± SDs. ### *p* < 0.001 vs. the NOR group; *** *p* < 0.001 vs. the CTL group.

## Data Availability

The data presented in this study are available upon request from the corresponding author.

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
