# Peer review of "The Anti-Inflammatory and Skin Barrier Function Recovery Effects of Schisandra chinensis in Mice with Atopic Dermatitis"

_medicina, 2023, doi:10.3390/medicina59071353_

Round 1
Reviewer 1 Report
The study aimed to test the anti-inflammatory and skin barrier function recovery effects of Schisandra chinensis in mice with atopic dermatitis.
The main aim of the study is not clear in the abstract.
The introdution is so brief.
The appropriate references were used but not and adequate. Please add some more recent and relevant references.
What were the standards or references for the procedures?
What was the sample size?
What was the method for determination of the sample size?
What were the statistical methods.
There are some sentences need reference(s). Please check totally.
The results were presented in good quality.
The discussion is good.
The conclusion is good.
The English used correct and readable. There are some minor errors. Please check.
The English used correct and readable. There are some minor errors. Please check.
Author Response
Response to Reviewer 1 Comments
Point 1: The main aim of the study is not clear in the abstract.
Response 1: The main research areas and additional investigation contents are separately described (L16-17).
Point 2: The introdution is so brief.
Response 2: Interest in S. chinensis as a raw material for cosmetics was mentioned along with references. (L37-40). Added content on natural products as supplementary alternatives to corticosteroids (L49-51)
Point 3: The appropriate references were used but not and adequate. Please add some more recent and relevant references.
Response 3: Older or less relevant references have been replaced and added new additional references. (L37-40 and others). In addition, changed to more appropriate references (L256-L258, L258-259, L264-265).
Point 4: What were the standards or references for the procedures?
Response 4: We induced dermatitis by slightly modifying the method of Martinez et al. [19], and conducted a preliminary study for this purpose. The method of topical treatment followed our standard method [18].
- Jang, S.K.; Kang, Y.H.; Kang, Y.T.; Oh, S.Y.; Kim, S.Y.; Lyu, J.H.; Kim, H.Y. The ethanol extract of Caragana sinica ameliorated skin lesions in mice with contact dermatitis. Pharmacognosy Magazine 2022, 18(77), 201-206. doi:10.4103/pm.pm_370_21
- Moosbrugger-Martinz, V.; Schmuth, M.; Dubrac, A Mouse Model for Atopic Dermatitis Using Topical Application of Vitamin D3 or of Its Analog MC903. Methods and Protocols 2017, 1559, 91-106. doi: 10.1007/978-1-4939-6786-5_8
Point 5: What was the sample size?
Response 5: The numbers of animals used in all groups are presented in the Materials and Methods section (L84-85).
Point 6: What was the method for determination of the sample size?
Response 6: We know that the presentation of the sample size determination method is recommended according to the ARRIVE standard. However, for animal studies, the minimum number of animals should be used according to the 3R rule. In our laboratory, 6 normal groups, 8 control groups, and 8 experimental groups were used to secure statistical significance.
Point 7: What were the statistical methods.
Response 7: Statistical analysis methods are described in the 2.11. section.
Point 8: There are some sentences need reference(s). Please check totally.
Response 8: We have added references to a few sentences (L256, L273 etc.).
Point 9: The English used correct and readable. There are some minor errors. Please check.
Response 9: We rechecked the entire manuscript and corrected a few sentences.
Correct spacing (L385), period addition (L427), semicolon addition (L430-431)
Others : Added information about research protocol approval.
Correction of reference order
Reviewer 2 Report
The manuscript is well structured, a good english but I have some questions:
you mentioned the epidermal changes but less about ceramides, filagrin, involucrin and their role in skin barier function which is impaired in AD; eosinophils and degranulated mastocyte in the infiltrate but less basophils and neutrophils
also Il13 is mentioned only in discussion;
the mice model of AD does not show any vesiculation lesions at clinical site,but has spongiosis and mycroabcesses in histopathologic view; eosinophils or mastocyte in the infiltrate.
As I noted up in the questionaire, consider that minor revision of english is neccessary.
Author Response
Response to Reviewer 2 Comments
Point 1: you mentioned the epidermal changes but less about ceramides, filagrin, involucrin and their role in skin barier function which is impaired in AD; eosinophils and degranulated mastocyte in the infiltrate but less basophils and neutrophils
Response 1: We added an explanation of our results by referring to studies on the effects of S. chinensis and its active ingredients on natural moisturizing factors (L222-229). We also added an explanation of our results by referring to studies on the effects of active components of S. chinensis on the infiltration of eosinophils and mast cells (L242-245).
Point 2: also Il13 is mentioned only in discussion;
Response 2: Although IL-13 plays a role along with IL-4 as a Th2 skewing cytokine, it was not the focus of this study. Therefore, it was deleted.
Point 3: the mice model of AD does not show any vesiculation lesions at clinical site, but has spongiosis and mycroabcesses in histopathologic view; eosinophils or mastocyte in the infiltrate.
Response 3: When contact dermatitis (CD) or AD is induced using chemicals such as DNFB, DNCB or oxazolone in experimental animals, skin thickness increases and epidermal hyperplasia tends to appear. In addition, scabs and hyperkeratosis due to exudate may be accompanied. However, if moderate or severe dermatitis is induced, epidermal detachment (eg. epidermolysis) occurs and the epidermal thickness does not increase. As shown in Figure 1Ab, repeated application of MC903 resulted in moderate to severe dermatitis lesions including induration and fissure. In our model, various phenomena caused by exudate, which are commonly seen in dermatitis caused by DNFB, tended to decrease and induration and fissure increased. Nevertheless, spongiotic changes as hallmarks of severe inflammation, disrupted the basal cell layer, an indistinct boundary between the epidermis and dermis, and superficial capillaries were shown in the histopathological observations (Figure 3Ab).
In another study using MC903, no vesiculation lesions were observed on the skin surface, but histopathology tended to show tissue destruction. Past studies in our laboratory also showed that the MC903 model was less likely to develop exudates or pustules than the DNFB model.
Others :
Added information about research protocol approval.
Correction of reference order
Round 2
Reviewer 1 Report
It can be accepted. Thanks